# The P2Y_2_ Receptor C-Terminal Tail Modulates but Is Dispensable for β-Arrestin Recruitment

**DOI:** 10.3390/ijms23073460

**Published:** 2022-03-22

**Authors:** Eline Pottie, Jolien Storme, Christophe P. Stove

**Affiliations:** Laboratory of Toxicology, Department of Bioanalysis, Faculty of Pharmaceutical Sciences, Ghent University, Campus Heymans, Ottergemsesteenweg 460, B-9000 Ghent, Belgium; eline.pottie@ugent.be (E.P.); jolien.storme@ugent.be (J.S.)

**Keywords:** P2Y_2_ receptor, β-arrestin2, β-arrestin1, nanoluciferase, live-cell reporter assay

## Abstract

The P2Y_2_ receptor (P2Y_2_R) is a G protein-coupled receptor that is activated by extracellular ATP and UTP, to a similar extent. This allows it to play roles in the cell’s response to the (increased) release of these nucleotides, e.g., in response to stress situations, including mechanical stress and oxygen deprivation. However, despite its involvement in important (patho)physiological processes, the intracellular signaling induced by the P2Y_2_R remains incompletely described. Therefore, this study implemented a NanoBiT^®^ functional complementation assay to shed more light on the recruitment of β-arrestins (βarr1 and βarr2) upon receptor activation. More specifically, upon determination of the optimal configuration in this assay system, the effect of different (receptor) residues/regions on βarr recruitment to the receptor in response to ATP or UTP was estimated. To this end, the linker was shortened, the C-terminal tail was truncated, and phosphorylatable residues in the third intracellular loop of the receptor were mutated, in either singly or multiply adapted constructs. The results showed that none of the introduced adaptations entirely abolished the recruitment of either βarr, although EC_50_ values differed and time-luminescence profiles appeared to be qualitatively altered. The results hint at the C-terminal tail modulating the interaction with βarr, while not being indispensable.

## 1. Introduction

P2Y receptors are G protein-coupled receptors (GPCRs) that are activated by extracellular nucleotides, either purines or pyrimidines. The P2Y_2_ receptor (P2Y_2_R) is one receptor within this class, and is mainly expressed in immune cells, epithelial and endothelial cells, kidney tubules, and osteoblasts. Endogenously, the P2Y_2_R is activated by both ATP (adenosine 5′-triphosphate) and UTP (uridine 5′-triphosphate) [1,2]. Physiologically, this receptor plays a role in, e.g., the regulation of the vascular tone and bone formation, in inflammation processes, and in the turnover of epithelial skin keratinocytes and urothelial bladder and ureter cells [3]. On the other hand, the P2Y_2_R is (presumably) involved in pathological processes, with examples in cardiovascular diseases, central nervous system disorders, and diseases of the airways, kidney, liver, and skin, inter alia [4]. The P2Y_2_R is the pharmacological target of diquafosol, an agonist approved in Korea and Japan for the treatment of dry eye disease, and agonists and antagonists are additionally suggested to be a valuable therapy in a range of other diseases [5,6,7].

Activation of GPCRs results in coupling to canonical G proteins, typically followed by a plethora of signaling events. Specifically, the P2Y_2_R primarily interacts with G_q_ upon receptor activation [1,5,8]. This interaction subsequently induces activation of phospholipase C-β, an enzyme responsible for the production of second messengers IP_3_ (inositol 1,4,5-triphosphate) and diacylglycerol through the hydrolysis of PIP_2_ (phosphatidylinositol-4,5-diphosphate). These second messengers then induce intracellular Ca^2+^ release and the activation of PKC (protein kinase C), respectively. Through activation of phospholipase A_2_ (PLA_2_), arachidonic acid can be released, with associated distinct signaling pathways [1,7]. Additionally, via interaction with α_v_β_3/5_ integrins, the P2Y_2_R can interact with G_0_ and G_12_ [8,9].

Besides the canonical G proteins, the signaling functions of GPCRs are additionally regulated through a different group of proteins, which are called arrestins. Arrestins were originally described as mediators of receptor desensitization, internalization, and intracellular receptor trafficking. Furthermore, they can act as scaffolding proteins, implicating a role as key regulators of a distinct set of signaling events, with functions in, e.g., cell growth, migration, and survival. Because of these diverse roles, arrestins can serve as potential therapeutic targets in various conditions, through various mechanisms [10,11]. The interaction of the GPCR with arrestin is thought to be mediated by either the receptor core, the phosphorylated C-terminal tail of the receptor, or a combination thereof, with phosphorylation patterns influencing the effect exerted by the arrestin [12,13,14].

The recruitment of β-arrestins to the P2Y_2_R and the exact mechanism(s) through which this occurs has only been scarcely explored [15,16,17]. However, therapeutic opportunities might lie in so-called biased agonism, in which GPCR binding to a ligand results in the preferential stimulation of a (subset of) pathway(s), while disfavoring others. Examples could be the selective activation or inhibition of β-arrestin recruitment, or functional selectivity between the different functions of β-arrestin, ideally leading to less adverse events [13,18,19]. Therefore, the aim of this study was to gain insight into the role of the C-terminal tail and the third intracellular loop (IL3) of the P2Y_2_R on the ability of the receptor to recruit either β-arrestin 1 (βarr1, arrestin 2) or β-arrestin 2 (βarr2, arrestin 3). More specifically, the receptor was adapted through truncation of the C-terminal tail or mutation of the serine and threonine (phosphorylatable) residues in IL3, and the ability of these mutant receptors to induce arrestin recruitment was assessed via cell-based assays employing functional complementation of a split nanoluciferase.

## 2. Materials and Methods

### 2.1. Chemicals and Reagents

Human Embryonic Kidney (HEK) 293T cells (passage 20) were kindly provided by Prof. O. De Wever (Laboratory of Experimental Cancer Research, Department of Radiation Oncology and Experimental Cancer Research, Ghent University Hospital, Belgium). The human P2Y_2_R construct (NM 002564.3, transcript variant 2 of the P2RY2 gene) was a kind gift from Prof. L. Erb (Department of Biochemistry, Life Sciences Center, University of Missouri, Columbia, MO USA). The plasmid containing the sequence of β-arrestin 1 was a kind gift of Dr. A. Chevigné (Immuno-Pharmacology and Interactomics, Department of Infections and Immunology, Luxembourg Institute of Health, Luxembourg). Dulbecco’s Modified Eagle’s Medium (DMEM, supplemented with GlutaMAX^®^), Phosphate-Buffered Saline (PBS), Fetal Bovine Serum (FBS), Penicillin/streptomycin, OptiMEM^TM^, Hank’s Balanced Salt Solution (HBSS), and EDTA were purchased from Thermo Fisher Scientific (Pittsburg, PA, USA). Reference agonists ATP and UTP, sodium azide, and poly-D-lysine hydrobromide were from Sigma Aldrich (Steinheim, Germany). Mouse anti-HA.11 Tag antibody (clone 16B12) and goat anti-mouse antibody labeled with Alexa Fluor 488 were purchased from BioLegend (San Diego, CA, USA). FuGENE^®^ HD transfection reagent and Nano-Glo^®^ Live Cell Reagent were from Promega (Madison, WI, USA).

### 2.2. Generation of the Plasmid Constructs in the NanoBiT^®^ System

Fusion constructs of the P2Y_2_R were generated in the NanoBiT^®^ system (Promega), with a peptide linker (‘GAQGNSGSSGGGGSGGGGSSG’) connecting the full-length receptor with one of the subunits of the split-NanoLuc^®^ luciferase (LgBiT or SmBiT), hence generating P2Y_2_R-LgBiT and P2Y_2_R-SmBiT. To exclude a role of the serine residues of the linker sequence in the recruitment of the arrestins, the linker was shortened to ‘GGGG’, generating a construct denoted as P2Y_2_R**L**-LgBiT. Subsequently, the C-terminus of the two P2Y_2_R constructs was truncated after proline 322, yielding constructs P2Y_2_R**Δ322**-LgBiT and P2Y_2_RL**Δ322**-LgBiT. In the next step, three serine/threonine residues—T232, S233, and S243—in IL3 of P2Y_2_RLΔ322 were mutated to alanine, yielding the mutated, truncated P2Y_2_R construct P2Y_2_RLΔ322-**AAA**-LgBiT. For each of the receptor constructs, an HA-tagged counterpart was cloned. The generation of the βarr1 and P2Y_2_R constructs, and the mutation and truncation of the latter, were performed via experimental conditions described previously [20], with specific primers, PCR conditions, and restriction enzymes provided in Appendix A. The generation of the βarr2 constructs in the NanoBiT^®^ system was reported previously [21]. The correctness of all generated constructs was verified via Sanger sequencing.

Figure 1 shows a schematic overview of the experimental setup outlined below, with panel A providing an overview of the C-terminally truncated residues (blue) and mutated IL3 residues (gold), panels B–E schematically depicting the generated constructs in the NanoBiT^®^ system, and panel F focusing on the mechanism of this latter system, in which P2Y_2_R activation leads to a luminescent readout upon functional complementation of the nanoluciferase enzyme fragments.

### 2.3. Routine Cell Culture Conditions and P2Y_2_R NanoBiT^®^ βarr1/2 Recruitment Assays in HEK 293T Cells

HEK 293T cells were routinely cultured in DMEM supplemented with GlutaMAX^®^, 10% heat-inactivated FBS, 100 IU/mL of penicillin, 100 µg/mL of streptomycin, and 0.25 µg/mL of amphotericin B in a humidified atmosphere, at 37 °C and 5% CO_2_. The protocol followed for the NanoBiT^®^ recruitment assays was largely the same as described before for the A_3_ adenosine receptor [20]. In brief, the routinely cultured cells are seeded in 6-well plates at a density of 500,000 cells per well. After overnight incubation, transient transfection takes place, employing FuGENE^®^ HD transfection reagent at a 3:1 FuGENE:DNA ratio, according to the manufacturer’s protocol. The transfection mixtures (in reduced serum medium, OptiMEM^TM^) consist of 1 µg of one of the (non-HA tagged) P2Y_2_R constructs and one of either the βarr1 or βarr2 constructs in the NanoBiT^®^ system, with 1.3 µg of pcDNA3.1. One day after transfection, the cells are reseeded into poly-D-lysine-coated 96-well plates, at a density of 50,000 cells per well, followed by another 24 h incubation before the readout takes place. The cells are washed twice with HBSS, after which 90 µL of HBSS is pipetted into each well. To this, 25 µL of Nano-Glo^®^ Live Cell Reagent (diluted 1/20 in Nano-Glo^®^ LCS Dilution buffer, according to the manufacturer’s protocol) is added, and the plate is transferred to the Tristar^2^ LB 942 multimode microplate reader (Berthold Technologies GmbH & Co., Bad Wildbad, Germany), in which the luminescence is monitored until stabilization of the signal. Upon reaching this equilibrium, 20 µL of 6.75× concentrated agonist solution (in HBSS) is added to each well, and luminescence values are registered (in real time) for the following 90 min. For all experiments, the final in-well concentrations ranged from 10 nM to 100 µM of agonist. Appropriate solvent controls were included.

### 2.4. Verification of P2Y_2_R Surface Expression

To verify the surface expression of the P2Y_2_R constructs and mutants linked to the components of the NanoBiT^®^ system, the presence of the HA-tag (N-terminally present in all HA-tagged versions of the constructs) at the cell surface was assessed via flow cytometry. HEK 293T cells were seeded in 24-well plates at a density of 80,000 cells per well. After overnight incubation, the cells were transfected with 275 ng of the (HA-tagged) receptor construct and 50 ng of transfection control (mTurquoise) in OptiMEM^TM^, employing a 3:1 FuGENE:DNA ratio (according to the manufacturer’s protocol). Twenty-four hours post transfection, the cells were detached, washed with flow cytometry buffer (consisting of PBS, 5% FBS, 2 mM of EDTA, and 2 mM NaN_3_), and incubated with 1 µL of mouse anti-HA.11 primary antibody for 30 min at 4 °C. After washing the cells (on ice) twice with flow cytometry buffer, and incubation at 4 °C for 30 min with 0.5 µL of secondary goat anti-mouse antibody labeled with Alexa Fluor 488, fluorescence was measured using the CytoFLEX flow-cytometer (Beckman Coulter Life Sciences, Brea, CA, USA). The fraction of cells with surface expression of the HA-tag was assessed via the fraction of FITC positive cells in the mTurquoise positive cells.

### 2.5. Data Analysis

Results of the flow cytometry analyses were processed using FlowJo version 10 (Ashland, OR, USA). The time-luminescence profiles in the NanoBiT^®^ recruitment assays were interpreted as described before in more detail, aiming to determine EC_50_ values as a measure of functional potency for the reference agonists for each of the generated constructs [23]. In brief, the time-luminescence profiles obtained in the NanoBiT^®^ recruitment assay were corrected for inter-well variability, and used for the calculation of AUC (area under the curve) values. The AUC of the appropriate solvent control sample was subtracted, and this value was subsequently used for the normalization and fitting of concentration-response curves in GraphPad Prism software (San Diego, CA, USA). A non-linear regression model (Hill Slope 1) was used for the for the calculation of EC_50_ values. The statistical significance of the differences between receptor mutants of the obtained EC_50_ values of the three independent experiments was estimated via Kruskal–Wallis analysis with post hoc Dunn’s test.

## 3. Results

### 3.1. Setup of a NanoBiT^®^ Bioassay to Monitor βarr Recruitment

To assess the recruitment of either βarr1 or βarr2 to the P2Y_2_R, the NanoLuc Binary Technology (NanoBiT^®^) system was employed. This system was specifically developed to monitor protein-protein interactions, and consists of two inactive split parts (the larger part, LgBiT, and the smaller part, SmBiT) of a nanoluciferase enzyme. In this specific case, one of the split fragments was fused to the C-terminus of the P2Y_2_R, and the second part to the N- or C-terminus of the respective βarr. Interaction of the concerned proteins leads to functional complementation of the enzyme fragments, which can be monitored through a luminescent readout in the presence of the enzyme’s substrate [24]. To obtain the highest possible sensitivity with the employed assay system, the optimal combination of P2Y_2_R and βarr2, each fused to either LgBiT or SmBiT, was determined. Figure 2 shows the obtained time-luminescence profile for each of the possible combinations, both in unstimulated cells (‘blank’, measured in cells to which solvent control was added, depicted in black) and in cells stimulated with 100 µM UTP (‘UTP’, time-luminescence profile provided in red). This latter concentration was selected because it is substantially higher than the reported EC_50_ values of UTP in the low micromolar range [25]. From the Figure, it is clear that the conditions in which the enzyme fragment is fused to the N-terminus of βarr2 yield the strongest relative increase in signal upon agonist addition, visible as the difference between the black curve (cells to which solvent control, ‘blank’, was added), and the red curve (cells stimulated with 100 µM UTP). These constructs with enzyme fragments at the N-terminus are ‘LgBiT-βarr2′ and ‘SmBiT-βarr2′, and are shown in panels C and D. As the combination of constructs shown in panel D yielded a slightly higher increase in signal (red profiles versus the control black profiles) than the combination of constructs shown in panel C, we opted to use the former, i.e., P2Y_2_R-LgBiT combined with SmBiT-βarr2, for further experiments. To eliminate as much as possible variability between the conditions with βarr1 and βarr2, the equivalent combination of constructs was applied for βarr1, i.e., P2Y_2_R-LgBiT and SmBiT-βarr1.

To assess the recruitment of βarr and thereby generate concentration-response curves, the obtained luminescence values need to be concentration-dependent. The results of this test, employing a concentration range (10 nM–100 µM) of the reference endogenous agonist UTP, for both βarr1 and βarr2, are shown in Figure 3. For both the recruitment of βarr1 or βarr2 to the P2Y_2_R, a clear concentration-dependent response was observed.

### 3.2. Surface Expression of the Generated P2Y_2_R Constructs in the NanoBiT^®^ System

Upon verification of the concentration dependence, certain adaptations were introduced into the P2Y_2_R-LgBiT construct to assess the influence on its ability to recruit βarr1 and βarr2. Hence, the following constructs were generated, with an increasing number of modifications: P2Y_2_R**L**-LgBiT (with a shorter linker, in which the phosphorylatable serine residues are removed); P2Y_2_R**Δ322**-LgBiT and P2Y_2_R**LΔ322**-LgBiT (in which the C-terminal tail of the GPCR is truncated); and P2Y_2_R**LΔ322**-**AAA**-LgBiT (in which three serine/threonine residues in IL3 are mutated to alanine). For optimal comparability of the results obtained with the different receptor constructs in the NanoBiT^®^ system, correct expression at the cell surface of the HA-tagged constructs was first confirmed via flow cytometry (Figure 4). Panel A shows the result of a control experiment where signals obtained for cells transfected with HA-tagged P2Y_2_R-LgBiT were compared to ‘negative control’ cells (non-transfected cells without antibody, with only a secondary (labeled) antibody, or with a primary and a secondary antibody). From this panel, it can be derived that the cells transfected with HA-tagged construct indeed gain a higher fluorescence intensity than the untransfected cells. Panel B then shows the assessment of the surface expression of the different HA-tagged P2Y_2_R constructs. To this end, the fluorescence (FITC) intensity for the cells that were positive for the mTurquoise transfection control (in contrast to the cells shown in panel A, which shows non-selected cells) is shown for each of the five generated constructs. As can be seen in panel B, the signals for all tested conditions largely overlap, indicating the similar surface expression of each of the generated HA-tagged P2Y_2_R-constructs in the NanoBiT^®^ system.

### 3.3. Assessment of βarr Recruitment by the Generated Constructs

For each of the generated (non-HA tagged) P2Y_2_R-LgBiT constructs, the ability to recruit either βarr1 or βarr2 upon stimulation with either of the endogenous agonists, UTP or ATP, was assessed. The obtained EC_50_ values, as a measure of the potency of the agonist to induce the recruitment of βarr1 or βarr2 to the (adapted) receptor constructs, are given in Table 1, and schematically represented in Figure 5. Focusing on the recruitment of βarr1 to the P2Y_2_R in the NanoBiT^®^ system, a first condition to be assessed was the effect of the presence of phosphorylatable amino acid residues in the sequence linking the receptor to the LgBiT fragment. When comparing the EC_50_ values obtained with P2Y_2_R-LgBiT and P2Y_2_R**L**-LgBiT, this influence appears to be of minor importance, as the values in the latter situation are only slightly higher than those in the former, with overlapping confidence intervals. The effect of the truncation of the receptor C-terminus is more outspoken, as assessed via P2Y_2_R**Δ322**-LgBiT and P2Y_2_R**LΔ322**-LgBiT. More specifically, the EC_50_ values obtained with the former construct are 2.4-fold higher than that of P2Y_2_R-LgBiT for UTP and 2.5-fold higher for ATP, and 4.75- and 3.5-fold, respectively, when comparing P2Y_2_R**LΔ322**-LgBiT with P2Y_2_R**L**-LgBiT. Supplementary mutation of the Ser/Thr residues in IL3 (P2Y_2_R**LΔ322**-**AAA**-LgBiT) induced a rather minimal additional change in the potency values of both UTP and ATP. Overall, the EC_50_ values of the different receptor constructs for UTP and ATP in the βarr1 assay span a relatively broad range, with 5.38- and 5.87-fold differences between the unmodified receptor construct and the P2Y_2_R**LΔ322**-**AAA** situation, respectively. However, all situations yielded EC_50_ values in the low micromolar range, confidence intervals mostly overlap, and only comparison of the unmodified construct with the maximally mutated construct (P2Y_2_R**LΔ322**-**AAA**) yielded statistically significant differences in EC_50_ value for ATP.

For the recruitment of βarr2, on the other hand, the obtained EC_50_ values for UTP and ATP lie more closely together for the different P2Y_2_R constructs, albeit still in the low micromolar range. More specifically, a 3.22-fold and 2.23-fold change (respectively) was observed between the EC_50_ values obtained for UTP and ATP using the P2Y_2_R construct versus the P2Y_2_R**LΔ322**-**AAA** construct. Similarly to βarr1 recruitment, the shortened linker was of rather small influence on the potency obtained with P2Y_2_R**L**-LgBiT *versus* P2Y_2_R-LgBiT. However, contrarily to βarr1, the EC_50_ values obtained for UTP and ATP with the former receptor construct were slightly lower than those obtained using the latter, although confidence intervals were still overlapping. The truncation of the receptor C-terminal tail, similarly to observations for βarr1 recruitment, resulted in increased EC_50_ values: the P2Y_2_R**Δ322**-LgBiT construct yielded a 1.7- and 1.4-fold higher EC_50_ value for UTP and ATP, respectively, than P2Y_2_R-LgBiT. Comparing P2Y_2_R**LΔ322**-LgBiT *versus* P2Y_2_R**L**-LgBiT, 3.4- and 2.6-fold increases in EC_50_ values were observed for UTP and ATP, respectively. In accordance with the observations for βarr1 recruitment, the additional mutations in IL3 did not yield markedly higher EC_50_ values for βarr2 recruitment. The Kruskal–Wallis analysis showed that only the EC_50_ values of the P2Y_2_R**L**-LgBiT and the maximally mutated P2Y_2_R**LΔ322**-**AAA**-LgBiT were statistically significantly different, for both agonists, as shown in Figure 5.

In addition to the numerical EC_50_ values, the continuous live cell readout also allows for the generation of real-time time-luminescence profiles. A closer look at these profiles revealed a remarkably different course for the curves obtained with full length P2Y_2_R constructs as compared to C-terminally truncated P2Y_2_R constructs. This is obvious from Figure 6, comparing the βarr1 (panels A–D) and βarr2 (panels E–H) recruitment to P2Y_2_R-LgBiT and P2Y_2_R**Δ322**-LgBiT (panels A versus B; or E versus F) upon stimulation with the reference agonist UTP. A more plateau-like shape is obtained for the former (wt) than for the latter (truncated) P2Y_2_R. This dissimilarity in profile course was also observed when comparing P2Y_2_R**L**-LgBiT and P2Y_2_R**LΔ322**-LgBiT (panels C versus D; or G versus H), or when focusing on ATP (for which the profiles are provided in Appendix A). The Figures reflect the least plateau-like profile for the truncated P2Y_2_R**Δ322**-LgBiT construct, with a stronger effect of this truncation on the profile of βarr2 than on βarr1, with both of the endogenous agonists.

## 4. Discussion

The P2Y_2_R responds endogenously to ATP and UTP, contributing to the effects of both the basal release of these nucleotides by the cells, and the reaction to their increased release in response to stress situations, such as mechanical stress, viral infection, oxygen deprivation, and apoptotic stimuli. This ability provides important roles for the P2Y_2_R in both physiological and pathological conditions, which are fulfilled through the interplay with different effector molecules. As a GPCR, the receptor mainly interacts with the G_q_ protein with additional coupling to G_0_ and G_12_ through integrins, resulting in a plethora of signaling events. Furthermore, β-arrestins function as scaffolding proteins, warranting receptor internalization, desensitization, and trafficking, and engaging (distinct) downstream molecules. Besides G protein and arrestin interactions, the P2Y_2_R can also participate in GPCR oligomerization and receptor cross-talk. Together, all these interactions contribute to the functional outcome of P2Y_2_R activation. Despite the (patho)physiologic relevance of this receptor, many of the described signaling interactions, and the exact mechanism behind them, remain to be explored in more depth. Therapeutic opportunities could lie in the exploration of ligand bias, in which the ligand is tailored to induce beneficial outcomes, while reducing on-target adverse events [1,5,7,8,9,19].

Focusing on the interaction of the P2Y_2_R with βarrs (βarr1 and βarr2), a rather limited number of studies is available in literature [15,16,17]. Older studies had investigated the mechanisms of receptor desensitization, internalization, and trafficking, and the influence of C-terminal truncation of the P2Y_2_R, or the mutation of specific C-terminal and IL3 residues on these processes [25,26,27,28,29]. Garrad et al. found progressive truncations of the murine P2Y_2_R C-terminal tail to influence the concentration-response curves of receptor desensitization and sequestration, while leaving Ca^2+^ mobilization unaltered [26]. Additionally, Flores et al. reported that mutation of three putative phosphorylation sites (S243, T344, and S356) decreased the efficacy of agonist-induced internalization and desensitization of the Ca^2+^ response [27]. Arrestins are now typically ascribed to contribute to GPCR internalization, signaling, desensitization, and trafficking [10,11,15,16]. Therefore, in this study, we investigated the molecular determinants of the interaction of βarr1 and βarr2 with the P2Y_2_R in the NanoBiT^®^ functional complementation assay, which allows for a real-time readout of luminescence generation in a live cell assay system. In all NanoBiT^®^ experiments, appropriate solvent controls were included to compensate for the possible nucleotide release because of the mechanical stress during experimental conditions [30].

In particular, the four possible combinations of the P2Y_2_R (C-terminally fused to either LgBiT or SmBiT) and βarr2 (N- or C-terminally fused to either LgBiT or SmBiT) were generated in order to select the optimal configuration and thereby the optimal sensitivity. This led to the selection of P2Y_2_R-LgBiT combined with SmBiT-βarr1/2 (Figure 2, panel D). The concentration-dependence of the recruitment of both βarr1 and βarr2 in the generated system was evaluated using the endogenous agonists, yielding EC_50_ values in the low micromolar range, corresponding with those reported in a Ca^2+^ mobilization assay [25]. Following successful evaluation of the assay setup, several adaptations were introduced in the initial P2Y_2_R-LgBiT construct. This yielded P2Y_2_R**L**-LgBiT, P2Y_2_R**Δ322**-LgBiT, P2Y_2_R**LΔ322**-LgBit, and P2Y_2_R**LΔ322**-**AAA**-LgBiT, in which L stands for an adjusted linker (removing Ser/Thr residues), Δ322 for receptor truncation after Pro 322, and AAA for mutation of T232, S233, and S243 to alanine. Each modification was introduced to assess the impact of the removal of specific phosphorylation sites on the recruitment of βarr1 and βarr2 in response to ATP and UTP. The surface expression of the HA-tagged P2Y_2_R constructs was determined to be comparable (Figure 4B), thereby ruling out the option of differential (or inadequate) surface expression as a confounding factor in the interpretation of the obtained NanoBiT^®^ results [31].

In literature, the linker sequence and used tag have been described to affect the arrestin recruitment in specific cases [31]. The comparison of P2Y_2_R-LgBiT with P2Y_2_R**L**-LgBiT allowed for the observation that the shortening and removal of potential phosphorylation sites in the linker did not markedly alter the obtained EC_50_ values with either agonist, as reflected by overlapping confidence intervals and the lack of statistical significance, excluding a major role of this effect in our results. The increased EC_50_ value of P2Y_2_R**LΔ322**-LgBiT versus P2Y_2_R**Δ322**-LgBiT leads to the hypothesis that the former construct, containing a shortened linker, may have a reduced flexibility or a less-ideal orientation to allow optimal functional complementation of the split-nanoluciferase enzyme, because of the truncated receptor.

Rather unexpectedly, all generated receptor constructs were still capable of recruiting both βarr1 and βarr2 upon activation by both UTP and ATP in the NanoBiT^®^ system (Figure 5 and Table 1). Based on the observation that the P2Y_2_R**Δ322**- and P2Y_2_R**LΔ322**-LgBiT constructs were still able to recruit both βarr1 and βarr2, we concluded that the C-terminal tail is dispensable for this interaction. However, the changes in EC_50_ values, and mostly the pronounced differences in activation profiles (Figure 6 and Appendix A), suggest that this tail does modulate the P2Y_2_R-βarr interaction in a certain manner. The different extent to which the values and the profiles of both ATP and UTP are influenced by the P2Y_2_R truncation, in the comparison of βarr1 and βarr2, may indicate a (slightly) different interaction between the tail and either arrestin, although it is not clear how or if the NanoBiT^®^ activation profiles correlate with the functioning of βarr1 and βarr2. The additional mutation of Ser/Thr residues did not lead to a pronounced additional effect. Although certain differences were observed between the obtained EC_50_ values, all potencies were still in the lower micromolar range. Garrad et al. reported a 30 times higher concentration of UTP to be required for the agonist-induced desensitization of the C-terminally truncated murine P2Y_2_R response, and these truncations decreased the receptor sequestration and influenced the time course [26]. Flores et al. found that mutating S243, T344, and S356 (the latter two residues residing in the C-terminal tail) caused a marked reduction in P2Y_2_R desensitization and agonist-induced internalization [27]. The use of murine P2Y_2_R and the monitoring of different functional outcomes complicate the comparison of these results to ours.

Available studies on the recruitment of βarr1 and βarr2 by the P2Y_2_R have resulted in apparently divergent results [15,16,17]. Hoffmann et al. reported a different recruitment pattern of bovine βarr1 and βarr2 to a human P2Y_2_R upon stimulation with UTP or ATP in transfected HEK cells, employing confocal microscopy and a FRET-based technique. More specifically, P2Y_2_R activation with UTP led to the equal translocation of βarr1 and βarr2, which would lead to classification of the receptor as a class B GPCR. On the other hand, ATP translocated βarr2 to a greater extent than βarr1, behavior that fits with class A GPCRs [17,32]. Contrarily, siRNA knockdown studies in rat mesenteric arterial and rat aortic smooth muscle cells found the knockdown of βarr1 to attenuate UTP-stimulated P2Y_2_R desensitization (and signaling in the latter cell line), whereas βarr2 knockdown did not [15,16]. Here, we used a maximally similar set-up, assessing recruitment of equivalent βarr1/2 constructs to the same P2Y_2_R construct, in the same cellular context, with the same read-out. Using this set-up, our results indicate a similar impact of both UTP and ATP in the two recruitment assays, albeit with different extents to which the EC_50_ values and profiles of βarr1 and βarr2 are affected by the introduced modifications to the P2Y_2_R—with more pronounced differences being observed for the EC_50_ values of βarr1. However, as mentioned above, the use of different cell types, expression levels, readout methods, and species of which the receptor and arrestin are derived, severely hampers the comparability of the results obtained in different studies.

Recent studies have shed more light on the molecular recognition of the GPCR by βarr molecules, and the resulting effect of that recruitment [12,13]. Latorraca et al. (2018) showed the ability of both the receptor core and the phosphorylated C-terminal tail of the GPCR to independently induce recruitment of βarr1, each through interactions with specific domains of the arrestin molecule [12]. Latorraca et al. (2020) also showed that different phosphorylation patterns of the receptor can favor different arrestin conformations, where phosphate-dependent conformations, not necessarily related to the number of phosphorylated residues, may select among diverse arrestin functions [13]. Our results can be considered to be in line with these conceptual findings, as the removal of the C-terminal tail and the phosphorylatable residues in IL3 did not abolish but rather altered the recruitment of the arrestins to the receptor. This hypothesis appears to be supported by the differential recruitment profiles obtained upon truncating the C-terminal tail. To what extent this results in an altered functionality of the arrestin is not clear at this point and may be the topic for future research. Furthermore, the diverging fold changes between the different mutants obtained with βarr1 and βarr2 might suggest that different motifs may be of importance for the recruitment of either.

## 5. Conclusions

In conclusion, a NanoBiT^®^ functional complementation assay was successfully set up and employed to assess the influence of different P2Y_2_R residues/regions on the recruitment of βarr1 and βarr2 in response to receptor activation by the endogenous ligands ATP and UTP. These regions/residues included the linker between the receptor and the NanoLuc fragment, the receptor’s C-terminal tail, and phosphorylatable residues in the P2Y_2_R IL3. From the interpretation of numerical EC_50_ values, it appeared that the truncation of the receptor’s C-terminus entailed the most notable effect. However, all of the generated constructs were still able to induce recruitment of either arrestin in response to receptor activation by either ligand, with overlapping confidence intervals. Remarkably, when looking at the time-luminescence profiles obtained with the receptor mutants, the qualitative ‘course’ of these profiles showed to be affected to different extents. Further research is warranted in order to explore if these altered profiles can be correlated with potentially altered functionalities.

## Figures and Tables

**Figure 1 ijms-23-03460-f001:**
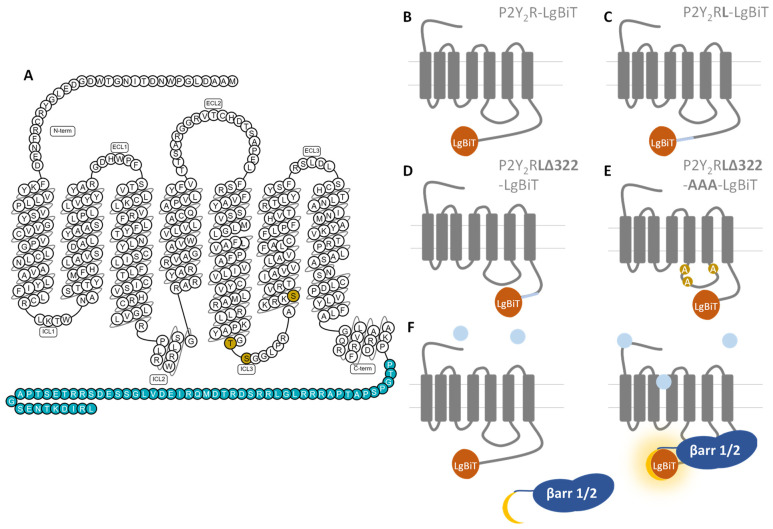
Schematic depiction of the experimental details of this manuscript. (**A**) P2Y_2_R snake plot (constructed on gpcrdb.org, last accessed 3 February 2022), with the residues removed in the truncated receptor constructs (**Δ322**) depicted in blue, and the mutated residues in IL3 (**AAA**) is shown in gold. Interaction with UTP supposedly occurs in the binding pocket generated by the upper parts of TM III, VI, and VII [22]. (**B**–**E**) Schematic overview of the generated receptor (mutant) constructs, with the exception of P2Y_2_R**Δ322**-LgBiT, which is C-terminally truncated but contains the original linker. (**F**) Schematic representation of the luminescence-based NanoBiT^®^ functional complementation assay, where agonist interaction with the receptor leads to recruitment of βarr1/2, functional complementation of the LgBiT and SmBiT enzyme fragments, and a luminescent signal.

**Figure 2 ijms-23-03460-f002:**
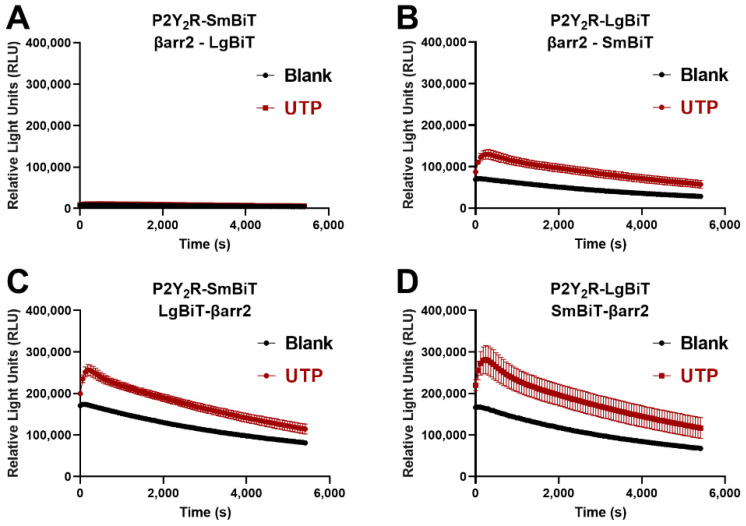
Selection of the optimal configuration of constructs of the P2Y_2_R and βarr2 in the NanoBiT^®^ system. Each panel shows one of the four possible combinations of the P2Y_2_R, C-terminally fused to either LgBiT or SmBiT, and βarr2, C-or N-terminally fused to LgBiT or SmBiT: (**A**) P2Y_2_R-SmBiT with βarr2-LgBiT, (**B**) P2Y_2_R-LgBiT with βarr2-SmBiT, (**C**) P2Y_2_R-SmBiT with LgBiT-βarr2, (**D**) P2Y_2_R-LgBiT with SmBiT-βarr2. The combination with the strongest increase in signal upon addition of 100 µM of UTP was selected, being P2Y_2_R-LgBiT with SmBiT-βarr2. Data from one representative experiment (of three independent experiments) are shown, performed in quadruplicate.

**Figure 3 ijms-23-03460-f003:**
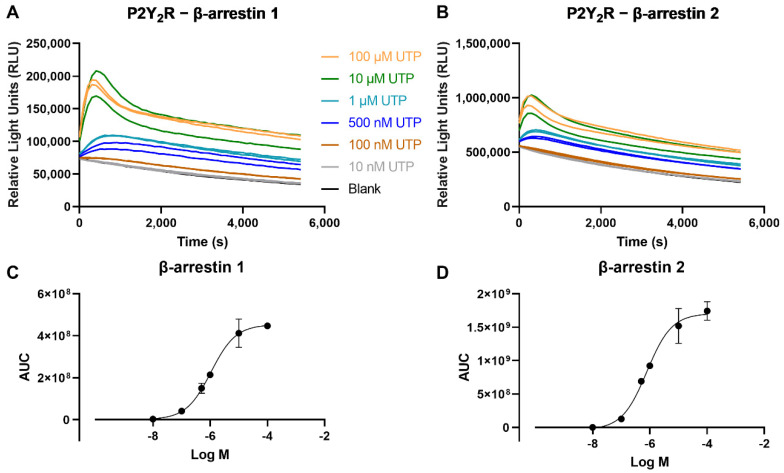
Time-luminescence curves of a concentration range of the reference agonist UTP in the NanoBiT^®^ system, monitoring the recruitment of (**A**) βarr1, or (**B**) βarr2 to the P2Y_2_R. Data shown are of one representative experiment out of three independent experiments, of which each condition was performed in duplicate. Panels (**C**,**D**) depict the corresponding sigmoidal concentration-response curves, with data points represented as mean ± SEM (standard error of the mean).

**Figure 4 ijms-23-03460-f004:**
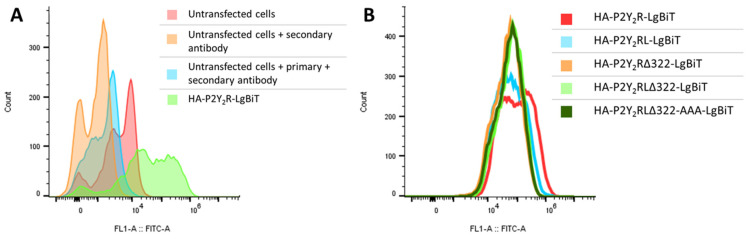
Outcome of the flow cytometry analysis to determine surface expression of the generated HA-tagged P2Y_2_R-constructs in the NanoBiT^®^ system. (**A**) Verification of the used protocol, comparison of control situations (untransfected cells with/without antibodies) and one of the generated constructs (in green). (**B**) Comparison of the fluorescence intensity between the different generated constructs after gating the cells that are positive for the transfection control mTurquoise. One representative experiment of three independent replicates is shown.

**Figure 5 ijms-23-03460-f005:**
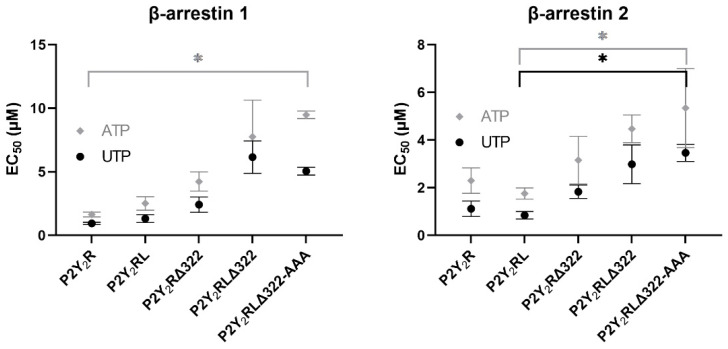
Visual representation of the obtained EC_50_ values, as a measure of the potency of either UTP or ATP to induce the recruitment of βarr1 or βarr2 to the (modified) P2Y_2_R constructs in the NanoBiT^®^ system. Data are ± SEM (standard error of the mean) of three independent experiments, each performed in duplicate. * *p* < 0.05, as assessed in a Kruskal–Wallis analysis with post hoc Dunn’s.

**Figure 6 ijms-23-03460-f006:**
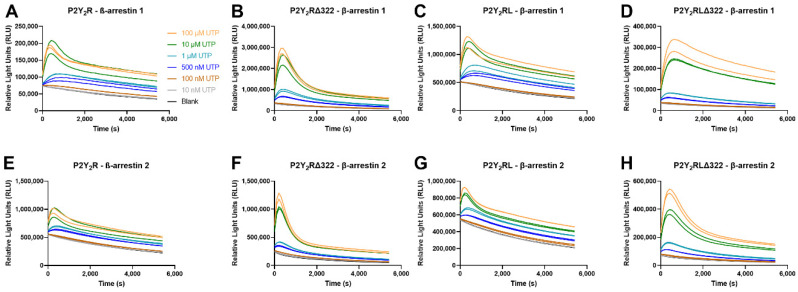
Time-luminescence profiles obtained in the NanoBiT^®^ assay by recruitment of βarr1 (panels **A**–**D**) or βarr2 (panels **E**–**H**) by the full length P2Y_2_R-LgBiT (panels **A**,**E**), the C-terminally truncated P2Y_2_R**Δ322**-LgBiT construct (**B**,**F**), P2Y_2_R**L**-LgBiT (**C**,**G**) or P2Y_2_R**LΔ322**-LgBiT, induced by reference agonist UTP. Data are from one representative experiment, out of three independent experiments.

**Table 1 ijms-23-03460-t001:** EC_50_ values, as a measure of the potency of either UTP or ATP to induce the recruitment of βarr1 or βarr2 to the (modified) P2Y_2_R constructs in the NanoBiT^®^ system. Data are from three independent experiments, each performed in duplicate. CI: 95% confidence interval.

Receptor Construct	β-Arrestin 1		β-Arrestin 2	
	EC_50_ UTP (µM, CI)	EC_50_ ATP (µM, CI)	EC_50_ UTP (µM, CI)	EC_50_ ATP (µM, CI)
P2Y_2_R-LgBiT	0.94 (0.52–1.68)	1.61 (0.94–2.84)	1.06 (0.46–2.55)	2.21 (1.20–4.44)
P2Y_2_R**L**-LgBiT	1.23 (0.55–0.29)	2.32 (0.86–7.34)	0.84 (0.55–1.26)	1.71 (0.97–3.23)
P2Y_2_R**Δ322**-LgBiT	2.24 (1.48–3.49)	4.04 (2.76–5.93)	1.82 (1.26–2.67)	2.98 (1.67–5.55)
P2Y_2_R**LΔ322**-LgBiT	5.82 (3.40–9.81)	8.10 (4.38–14.5)	2.83 (1.79–4.58)	4.42 (2.44–8.01)
P2Y_2_R**LΔ322**-**AAA**-LgBiT	5.04 (4.28–5.91)	9.46 (7.40–12.0)	3.43 (2.28–5.24)	4.93 (2.92–8.22)

## Data Availability

Data available upon reasonable request.

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
