# Peer review of "The P2Y2 Receptor C-Terminal Tail Modulates but Is Dispensable for β-Arrestin Recruitment"

_ijms, 2022, doi:10.3390/ijms23073460_

Round 1

Reviewer 1 Report

See attachment.

Reviewer 2 Report

The manuscript by Pottey et al investigates the role of C-terminal on the P2Y2 receptor for recruitment if beta-arrestin.  The authors have used NanoBit functional assay which suggests that C-terminal tail modulates the interaction of P2Y2 receptor with beta -arrestin but is not dispensable.

The manuscript is well written, however I have concerns outlined below

  1. Page 4 line 174, please justify dose used.
  2. Fig 1: What is the blank used? The values of blank are different in all panels. Please comment on this. What do the red lines indicate? Also, it is not required to box the optimum construct. Furthermore, signals are apparent in 3 panels so why authors decided to go for lgBit.
  3. Page 6 line 203, Does modifications induce any functional consequences in the receptor expression?
  4. Fig 3: Panel a: tagged construct shows more positive expression as indicated in flow and some overlay with negatives. However the intensity shows complete overlap which does not match with the real time flow recording. I am assuming the green curve in panel B should be shifted on the right?
  5. Table 1: The authors should provide statistics for EC50. looks like EC50 is increases for both stimulations for both beta arrestins. These results should be clearly mentioned and discussed and why authors describe some of data as minor importance.
  6. The authors try to elucidate the mechanism but the conclusion appears premature with limited data and repetition of results.
